# The Molecular Landscape of Hürthle Cell Thyroid Cancer Is Associated with Altered Mitochondrial Function—A Comprehensive Review

**DOI:** 10.3390/cells9071570

**Published:** 2020-06-27

**Authors:** Sonam Kumari, Ruth Adewale, Joanna Klubo-Gwiezdzinska

**Affiliations:** Metabolic Diseases Branch, National Institute of Diabetes and Digestive and Kidney Diseases, National Institutes of Health, Bethesda, MD 20892, USA; sonam.kumari@nih.gov (S.K.); ruth.adewale@nih.gov (R.A.)

**Keywords:** mitochondria, metabolism, Hürthle cell, oxidative phosphorylation, thyroid cancer, oncogenes

## Abstract

Hürthle cell thyroid carcinoma (HTC) accounts for 3–5% of all thyroid malignancies. Widely invasive HTC is characterized by poor prognosis and limited responsiveness to standard therapy with radioiodine. The molecular landscape of HTC is significantly different from the genetic signature seen in other forms of thyroid cancer. We performed a comprehensive literature review on the association between the molecular features of HTC and cancer metabolism. We searched the Pubmed, Embase, and Medline databases for clinical and translational studies published between 1980 and 2020 in English, coupling “HTC” with the following keywords: “genomic analysis”, “mutations”, “exome sequencing”, “molecular”, “mitochondria”, “metabolism”, “oxidative phosphorylation”, “glycolysis”, “oxidative stress”, “reactive oxygen species”, and “oncogenes”. HTC is characterized by frequent complex I mitochondrial DNA mutations as early clonal events. This genetic signature is associated with the abundance of malfunctioning mitochondria in cancer cells. HTC relies predominantly on aerobic glycolysis as a source of energy production, as oxidative phosphorylation-related genes are downregulated. The enhanced glucose utilization by HTC is used for diagnostic purposes in the clinical setting for the detection of metastases by fluorodeoxyglucose positron emission tomography (FGD-PET/CT) imaging. A comprehensive metabolomic profiling of HTC in association with its molecular landscape might be necessary for the implementation of tumor-specific therapeutic approaches.

## 1. Introduction

Differentiated thyroid cancer (DTC) arises from the follicular cells of the thyroid gland and is the fifth most common malignancy in women [1,2,3]. Based on histologic phenotypes, DTC is divided into three major groups: (1) the most common papillary thyroid cancer; (2) follicular thyroid cancer; (3) Hürthle cell thyroid cancer (HTC). HTC accounts for 3–5% of DTC and is characterized by its unique clinical and biological behavior [4,5,6,7]. Recent advances in elucidating the molecular landscape of HTC revealed that it is distinct from other DTCs, showing that it is not a subtype of follicular thyroid cancer, as previously believed [5,6]. Hürthle cells are large oxyphilic cells characterized by abundant mitochondria, prominent nucleoli, and loss of cell polarity [8,9,10]. The mitochondrial content results in an eosinophilic granular cytoplasm upon staining with hematoxylin and eosin. These properties of oxyphilic (i.e., oncocytic) cells were first thoroughly described in 1907 by Theodor Langhans [11]. Interestingly, oxyphilic cells can be present not only in cancer but also in benign thyroid conditions such as Hashimoto thyroiditis, multinodular goiter, and adenoma [8,9,10]. The differentiation between a benign thyroid Hürthle cell adenoma and HTC cannot be made solely on the basis of the cell morphology derived from the cytology material obtained from the fine needle aspiration biopsy (FNAB) [4]. Most often, ultrasonography reports and indeterminate cytological samples cannot provide a substantial identification of the pathological condition [10]. In such situations, final diagnoses can be obtained after partial or total thyroidectomy. HTC is characterized by a capsular and vascular invasion, while benign Hürthle cell adenoma lacks invasive features [4]. Thus, a detailed histopathological characterization is required in order to distinguish between benign Hürthle cell adenoma and HTC [12]. However, the inability to predict from FNAB specimens which Hürthle cell neoplasm is malignant has led to surgical over-treatment [4,13]. Therefore, there has been a significant effort in recent years to enhance the diagnostic accuracy of FNAB by combining it with the analysis of the molecular signature of thyroid nodules [13]. Currently available molecular tests, such as the Afirma Gene Sequencing Classifier (GSC; Veracyte, South San Francisco, CA, USA) and ThyroSeq v3.0 (test developed by Molecular and Genomic Pathology laboratory at The University of Pittsburgh Medical Center, Pittsburgh, PA, USA) have incorporated testing methodology that evaluates chromosomal loss and mitochondrial DNA mutations, which are typically seen in HTC [8,14,15]. The main distinction between benign Hürthle cell adenoma and HTC is based on copy number alterations and a nearly complete genome haploidization specific to HTC, as identified by ThyroSeq v3.0 [16]. The benign call rate is another measure used to compare Hürthle cell adenomas and HTC using GSC, which combines next generation sequencing with machine learning tools. As a result, an increasing number of indeterminate nodules have been determined as benign in nature [17,18]. Consequently, this has led to a significant improvement in the specificity and positive predictive value of the test, with the predicted ability to avoid up to 60% of unnecessary surgeries for benign conditions [8]. The remaining 40% is a result of the overlap between the benign and malignant Hürthle cell neoplasms that still exists and refers to common somatic mutations, such as variants of the *RAS* gene, associated with both benign and malignant thyroid tumors. The genetic information of the patients who test positive, combined with the clinical data and imaging techniques, can pave the way for individualized therapy [19].

HTC is further characterized based on the extent of vascular invasion—tumors with <4 foci of vascular invasion are categorized as minimally invasive, while tumors with ≥4 foci are categorized as widely invasive. Moreover, low-risk HTC is characterized by a different molecular profile compared with high-risk HTC, with the latter characterized by a significantly higher mutation burden, frequent loss of heterozygosity, and mitochondrial DNA mutations affecting complex I involved in the electron transport chain [5,6]. While minimally invasive HTC is characterized by an excellent prognosis, widely invasive HTC is associated with a high metastatic potential and high mortality [20,21,22,23,24]. The latter is partially due to the unresponsiveness of HTC to standard therapy with radioactive iodine or to chemotherapeutic agents [25,26,27,28,29]. It is critical to better understand the molecular landscape, biology, and metabolism of these tumors in order to find novel effective therapeutic approaches. Therefore, the goal of this study was to provide a comprehensive review of the molecular landscape of HTC, particularly the widely invasive form of HTC. 

## 2. Search Strategy and Selection Criteria 

We performed a thorough literature review in PubMed, Embase, and MEDLINE. Data were reviewed from full-length articles, including translational in vitro and in vivo studies, clinical case reports, case series, observational retrospective and prospective studies, systematic reviews, and meta-analyses, published between January 1, 1980 and March 31, 2020 in English. Search terms included HTC coupled with the following keywords: “exome sequencing”, “genomic analysis”, “mutations”, “molecular”, “mitochondria”, “oxidative phosphorylation”, “metabolism”, “glycolysis”, “reactive oxygen species”, “oxidative stress”, and “oncogenes”. The final reference list was created based on the originality of the articles as well as their relevance to the broad scope of this review. 

## 3. Genetic Alterations in HTC

Widely invasive HTC is characterized by a significantly—on average, six-fold higher— mutational burden than other forms of DTC [6,11]. This mutational burden resembles more aggressive tumors such as glioblastoma multiforme or ovarian cancer [11]. In contrast, minimally invasive HTC is characterized by a low mutation frequency (0.4 non-silent mutations per megabase), which is similar to more indolent forms of DTC such as classic papillary thyroid cancer [5]. 

The most common genetic alterations observed in HTC include pathogenic variants in the genes associated with abnormal protein translation, such as *EIF1AX* (eukaryotic translation initiation factor 1A X-linked), *MADCAM1* (mucosal vascular addressing cell adhesion molecule 1), or *DAXX* (death domain associated protein). *EIF1AX* is mainly involved in the preinitiation complex during translation. It is considered to be one of the genes involved in tumor initiation [30]. Apart from HTC, this gene is present in a highly mutated form in uveal melanomas [31], as well as in about 1% of papillary thyroid cancers [32]. The upregulation of *MADCAM1*, which is important for adhesion processes, can also lead to increased protein translation and phosphorylation of AKT, resulting in enhanced proliferation [33]. *DAXX* pathogenic variants act as potential driver mutation candidates for HTC tumorigenesis, as they lead to the altered transcriptional activities of various transcription factors [5,34]. These data suggest that the dysregulation of translation is highly significant for HTC’s pathogenesis [6]. 

Other pathogenic variants commonly observed in HTC lead to enhanced cell proliferation. These include pathogenic variants in the *RAS* family of oncogenes that are involved in controlling cell division, as well as mutations in the negative regulator of RAS pathway–*NF1* (neurofibromatosis type 1) and *CDKN1A* (cyclin dependent kinase inhibitor 1A), which is involved in cell cycle regulation [32,35,36,37]. In addition, commonly mutated *ATXN1* (antioxidant protein 1 homolog), involved in the regulation of the cell cycle and oxidative stress, may play a role in HTC oncogenesis [38]. Another important mutation leading to enhanced HTC tumorigenesis is observed in the *TP53* gene [5]. The *TERT* (telomerase reverse transcriptase) promoter mutations observed in HTC are essential for the immortalization of transformed cells [5,30,39]. These mutations are also seen in forms of thyroid cancer other than HTC. 

Interestingly, HTC is also commonly associated with pathogenic variants leading to altered cytoskeleton dynamics, such as mutations in *UBXN11* (UBX domain protein 11) [6]. Cytoskeleton abnormalities may facilitate metastatic potential and resistance to therapy [32] (Figure 1). Moreover, the somatic *GRIM-19* mutation which has been observed in HTC is one of the mutations of 19p13 which is associated with cell proliferation, apoptosis, and mitochondrial metabolism [40]. Maximo et al. discovered that germline *GRIM-19* pathogenic variants might be associated with familial forms of HTCs [41].

To summarize, the most common somatic mutations in HTCs are associated with aberrant signaling, leading to enhanced proliferation, abnormal protein translation, and altered cytoskeletal dynamics (Figure 1). 

## 4. LOH and Activation of AKT/mTOR Signaling

Another landmark of HTC, clearly distinguishing it from other types of thyroid cancer, is the widespread loss of heterozygosity (LOH) [5,6,11,42] (Figure 1). This feature is clearly associated with the invasive forms of primary tumors, characterized by a high recurrence rate and metastatic potential. 

A study utilizing the Cancer Genome Atlas (TCGA) database for the pan-cancer analysis of LOH suggested that, apart from HTC, sarcoma and glioblastoma multiforme were the only cancers to have more than 0.6 of the genome alterations caused by LOH. This strongly underscores the relevance of LOH in HTCs (Figure 1). The level of uniparental disomy (UPD) found in certain HTCs was remarkably high and, in certain cases, affected the complete genome. This widespread LOH is associated with the inactivation of several tumor suppressor genes [43]. Interestingly, tumors with high LOH can have enhanced genetic instabilities and tend to be activated with cyclin-dependent kinase signaling [6].

Recent studies also revealed whole-chromosomal duplication appearing in a non-random pattern, particularly in chromosomes 5 and 7, in HTC. Interestingly, this phenomenon is associated with the overactivation of the PI3K/AKT/mTOR pathway, along with the RAS/RAF/MAPK signaling pathways (Figure 1), leading to enhanced cell proliferation and reduced apoptosis [6]. 

## 5. Distinct Metabolic Profile of Cancer Cells

Cancer cells are distinct from normal cells due to their uncontrolled cell division phenotype. They require abundant amounts of energy to fulfill their accelerated growth and proliferation demands. The classic “Warburg effect” was introduced by Nobel Prize Laureate Otto Warburg in the 1920s. He proposed that cancer cells preferentially use glycolysis as a source of energy production despite an abundant supply of oxygen [44]. This was a striking discovery as glycolysis is not as efficient as oxidative phosphorylation (OXPHOS) in generating ATP [45]. This metabolic shift from mitochondrial respiration to fermentation is also called “aerobic glycolysis”. The major features associated with this process are elevated glucose uptake and accelerated lactate secretion. The enhanced lactate production results in an acidic tumor microenvironment, triggering neovascularization. The reverse Warburg effect is a complementary and newly identified phenomenon, which is also referred to as metabolic coupling. During this process, a close association occurs between the different populations of tumor and stromal cells (i.e., fibroblasts). As a result, the stromal fibroblasts and a subpopulation of tumor cells utilize aerobic glycolysis to generate lactate, which is distributed to the surrounding cancer cells as a nutrient-substrate, facilitating the TCA cycle. The increased ATP production resulting from this process is required for anabolic growth. Interestingly, this kind of metabolism enhances chemoresistance and leads to the reduced effectiveness of therapeutic strategies [46]. 

## 6. Mitochondrial DNA Mutations in HTC Leading to Decreased OXPHOS

One of the key elements in the molecular landscape of HTC, differentiating it from other forms of thyroid cancer, is the recurrent somatic loss of function and missense mitochondrial DNA (mtDNA) mutations in genes encoding subunits of complex I of the electron transport chain [5,6,11,42]. Notably, alterations in mitochondrial function and oxidative phosphorylation have been associated with widely invasive HTC [5,6]. 

It has been established that HTCs have excessive mitochondria as well as mitochondrial abnormalities to a much greater extent than has been observed in other forms of DTC. Approximately 71.4% of HTCs harbor non-synonymous mtDNA mutations and 36.7% are characterized by loss-of-function mtDNA variants, which clearly demonstrates the importance of mitochondrial abnormalities in HTCs [6]. 

The aberrant mitochondrial function observed in HTC could promote the shift from oxidative phosphorylation (OXPHOS) to aerobic glycolysis as the predominant source of ATP production [6]. It still needs to be established whether complex I mutations as well as the near-haploid state emerge at the same time in the course of cancer progression or whether they originate independently. It has been reported that mtDNA aberrations are capable of inducing epigenetic modifications within the nuclear genome, which might be associated with tumor formation [47,48]. 

The HTC expression profile patterns revealed that widely invasive HTC is characterized by a significant downregulation of the genes involved in OXPHOS, particularly those involved in the electron transfer chain, such as MT-CO1, MT-ATP6, MT-CO2, MT-CO3, MT-ND2, MT-ND4, MT-CYB, SDHA, COX6A1, COX5A, COX7A1, COX7B, CYCS, ATP5B, and UQCRQ [6]. The complex I mutations can occur in the ND1, ND2, ND3, ND4, ND5, ND6, and ND4L regions of the mitochondrial genome [49]. HTC harbors near-specific aberrations in ND2 and ND4 complex I regions. In contrast, mutations in the ND1, ND3, ND5, ND6, and ND4L regions of complex I have been found in other types of cancer, such as prostate cancer [50], pancreatic cancer [51], colon cancer [52,53], bladder cancer [54], breast cancer [55], and medulloblastoma [56].

Savagner et al. found that ATP synthesis was markedly decreased in Hürthle cell tumors in comparison to controls, indicating that the OXPHOS coupling deficiency might be responsible for the mitochondrial hyperplasia observed in oxyphilic thyroid tumors [57]. 

These observations were further supported by functional in vitro studies of the XTC.UC1 thyroid oncocytic (Hürthle cancer) cell line. In contrast to non-oncocytic cell lines, XTC.UC1 could not survive in media with galactose as the major nutrient. In this condition, cells are forced to meet their metabolic demands only through mitochondrial respiration. However, the mitochondrial respiration rate is significantly diminished in XTC.UC1 cells. These cells are also characterized by a significant reduction in the enzymatic activity of complex I as well as complex III of the mitochondrial respiration chain and increased ROS formation when compared to the control cells [58]. These observations were further validated in osteosarcoma derived transmitochondrial cell hybrids, also known as cybrids, which carried the mtDNA of XTC.UC1. The cybrid clones of XTC.UC1 displayed decreased ATP and reduced viability in a galactose-containing medium [58].

In contrast, Stankov et al. demonstrated that XTC.UC1 cells had elevated complex I and II activity but reduced activity in complex III. They were mostly dependent on OXPHOS for energy production and also formed excessive amounts of ROS [59]. The discrepancy in the results of the above-mentioned studies could be associated with different in vitro conditions and different methods of measuring mitochondrial respiration [59].

It is widely accepted that aberrant mitochondrial respiration and dysregulated mitochondrial function can result in increased oxidative stress, which can further lead to oncogenesis. Therefore, as Hürthle cells have abundant mitochondria, there was a recent attempt to investigate the association between the genes involved in the oxidative stress response and HTC. HTCs demonstrate increased production of reactive oxygen species (ROS), which enhances oxidative stress. These events further induce the oncogenic signaling pathways causing the malignant transformation of cells and resistance to several therapeutic drugs and radiation therapies [6]. Moreover, *NFE2L2* (nuclear factor erythroid 2-related factor 2) and *KEAP1* (kelch-like ECH-associated protein 1) mutations have been identified in samples derived from patients with HTC [5,6]. These alterations might be of importance, as *NFE2L2*, which is negatively regulated by *KEAP1*, enhances survival after cellular damage [60,61,62,63]. Moreover, there are no significant changes in the anti-oxidative stress machinery in HTC. Krhin et al. showed no direct association between the expression of antioxidant genes, including *GPX1* (glutathione peroxidase 1), *GSTP1* (glutathione-S-transferase P1), *GSTT1* (glutathione-S-transferase T1), *GSTM1* (glutathione-S-transferase M1), *SOD2* (superoxide dismutase 2), and *CAT* (catalase), and the development of HTC. Interestingly, the study suggested that the *GPX1* Pro198Leu polymorphism might be associated with the risk of HTC [64]. However, these findings need to be validated within a larger population. The dysregulated mitochondrial function leading to excessive ROS production and downregulated OXPHOS which has been observed in HTC is associated with the compensatory upregulation of aerobic glycolysis [65,66,67].

## 7. Enhanced Glycolysis in HTC

Phosphatidylinositide 3 kinases (PI3Ks), along with AKT and mTOR (downstream target of AKT), play a fundamental role in glycolysis and the regulation of glucose homeostasis by modulating glucose uptake and insulin action [68]. Overactive PI3K/AKT/mTOR signaling in HTC is associated with increased glycolysis as a major source of energy production (Figure 1). It has also been reported that in the thyroid gland and the pancreas, the PI3K-AKT pathway plays a crucial role in Glut-1 translocation from the cytoplasm to the plasma membrane [69,70].

Consistent with these observations, there is evidence for the upregulation of the genes involved in glycolysis in HTC. Kim et al. reported an overexpression of hexokinase II—the rate-limiting first enzyme in the glycolysis pathway. In addition, the glucose transporter GLUT1 and monocarboxylate transporter MCT4, which is responsible for lactate release from the cell, were significantly upregulated in HTC in comparison with benign thyroid tumors [71]. Moreover, the expression of hexokinase II was correlated with tumor size and elevated MCT4 expression was associated with the extrathyroidal extension of HTC. Additionally, carbonic anhydrase IX (CAIX), a cell pH regulator involved in the Warburg effect, has been found to be associated with vascular invasion of HTC [71]. The transcription factor HIF-1α (hypoxia inducible factor- 1α) is known to be strongly associated with increased glucose metabolism and angiogenesis in cancer [72,73,74,75,76]. Studies have reported expression of VEGF, a target gene of HIF-1α, in benign as well as malignant Hürthle cell tumors, suggesting the activation of HIF-1α pathway in these lesions [77]. So far, there has been no concrete evidence of the direct role of HIF-1α in promoting HTC tumorigenesis. The enhanced glucose uptake and increased glycolysis of HTC cells has been translated to the clinical setting and used for diagnostic and prognostic purposes through functional imaging relying on radiolabeled glucose uptake in HTC metastases. 

Fluorodeoxyglucose (FDG) is an analog of glucose that is widely utilized for PET/CT scans. In cancer tissues with accelerated glucose metabolism, there is a high uptake of ^18^FDG. HTCs are characterized by enhanced glucose uptake and reduced iodine uptake due to the decreased expression of the sodium-iodide symporter. Therefore, the sensitivity of FDG-PET/CT scanning in the detection of HTC metastases is significantly higher when compared with the gold standard in the imaging of DTC—diagnostic radioactive iodine whole body scanning [4,47,78,79]. 

## 8. Targeted Therapies for HTC

The various signaling pathways and metabolic alterations associated with HTC provide multiple potential therapeutic targets. The current standard therapy for radioactive iodine non-avid HTC consists of treatment with tyrosine kinase inhibitors (TKIs). There are two Food and Drug Administration (FDA)-approved agents for the management of metastatic progressive HTC—lenvatinib and sorafenib [80]. However, the efficacy of lenvatinib and sorafenib in HTC is limited. Therefore, there is a need for new therapeutic approaches targeting the overactive oncogenic signaling pathways as well as altered metabolism (Figure 2). Given the high mutation burden associated with increased immunogenicity, immunotherapy with immune checkpoint inhibitors has been utilized for advanced and metastatic HTCs. The monoclonal antibodies against programmed cell death protein 1 (PD-1), PD ligand 1 (PD-L1), and cytotoxic T lymphocyte associated antigen 4 (CTLA-4), have been utilized either as monotherapies or combination therapies [81]. Clinical trials which are focused on combination therapies for HTC include targeting the immunological landscape of HTC along with tumor angiogenesis, tyrosine kinase receptors, and overactive mTOR signaling. Motesanib (AMG706) is utilized to block tumor angiogenesis via the inhibition of vascular endothelial growth factors (VEGF) 1–3 as well as antagonizing platelet-derived growth factor receptor (PDGFR) and c-KIT signaling in metastatic thyroid cancer (NCT00121628, Phase II). Among combination therapies with TKIs and mTOR inhibitors, there is a clinical trial utilizing sorafenib tosylate alone or in combination with the mTOR inhibitor everolimus (NCT02143726, Phase II). Other ongoing clinical trials include a combination of the VEGF inhibitor cediranib maleate and immunomodulatory agent lenalidomide, which activates T and natural killer (NK) cells (NCT01208051, Phase I/II), (Figure 2). Some other trials utilizing TKIs and immunomodulatory agents are implementing a combination of cabozantinib and checkpoint inhibitors against PD1 (nivolumab) and CTLA4 (ipilimumab) (NCT03914300, Phase II) and a combination of lenvatinib with anti-PD1 monoclonal antibody (pembrolizumab) (NCT02973997, Phase II), (Figure 2). There is also an active clinical trial targeting the epigenetic regulation of the tumor with a histone deacetylase inhibitor (romidepsin) (NCT00098813, Phase II), (Figure 2).

## 9. Summary and Conclusions 

HTC is distinct from other cancers, as demonstrated by the extensive mtDNA mutations as well as the whole chromosome losses. These genetic alterations lead to decreased oxidative phosphorylation, enhanced aerobic glycolysis, and oxidative stress, as well as the overactivation of the PI3K/AKT/mTOR and RAS/RAF/MEK/ERK signaling pathways. Moreover, high-risk HTC is characterized by a significant mutation burden and aneuploidy, which may result in the enhanced immunogenicity of the tumor and consequently a better response to immunotherapy [82]. This distinct molecular landscape may form a basis for novel therapeutic approaches involving combination therapies affecting overactive signaling pathways, cancer metabolism, and immune landscape, as currently available treatments with radioactive iodine and FDA-approved TKIs are not effective. It is worthwhile to speculate that targeting overactive signaling pathways with TKIs and/or mTOR inhibitors in combination with small molecules blocking glucose transporters (GLUTs) or the enzymes involved in overactive aerobic glycolysis, such as hexokinase inhibitors, along with targeting the tumor microenvironment by immunotherapy, might be an effective strategy for novel clinical trials. HTC therapy is an evolving field and the recent findings described in this review may form a basis for identifying effective strategies which can be exploited to prolong the survival rate of patients with HTC.

## Figures and Tables

**Figure 1 cells-09-01570-f001:**
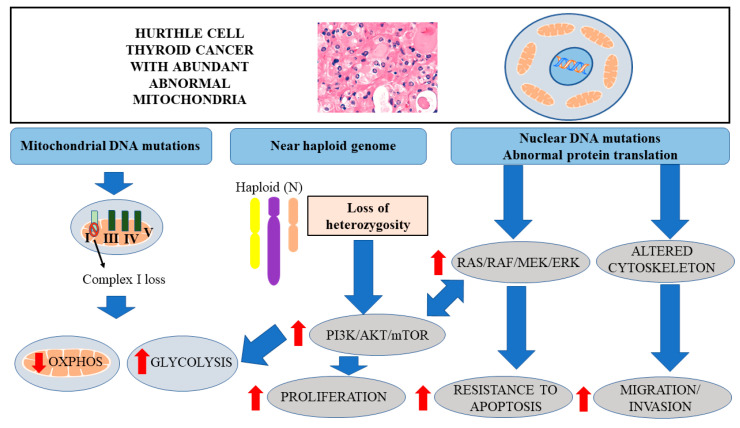
Diagram representing the role of mitochondrial mutations, near haploid genome and somatic mutations in Hürthle cell thyroid cancer development. Mitochondrial mutations, leading to the inactivation of complex I of the respiratory chain, are associated with the decreased OXPHOS and increased glycolysis. Near haploid genome, associated with the inactivation of several tumor suppressors, leads to the overactivation of PI3K/AKT/mTOR signaling pathway, enhancing cell proliferation and glycolysis. Somatic mutations in nuclear DNA are associated with abnormal protein translation, overactivation of PI3K/AKT/mTOR and RAS/RAF/MEK/ERK signaling pathways and altered cytoskeleton, leading to enhanced proliferation, resistance to apoptosis, and metastatic potential.

**Figure 2 cells-09-01570-f002:**
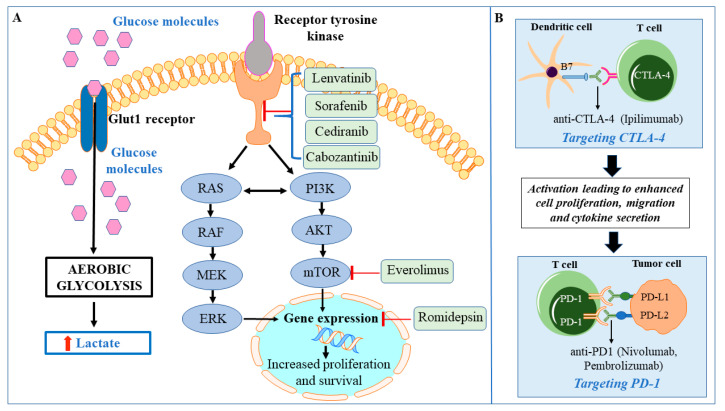
Model depicting the drug targets in Hürthle cell thyroid cancer that are being exploited in the ongoing clinical trials. (**A**) The inhibitory molecules acting upon various oncogenic pathways are shown in the green box. (**B**) Diagrammatic representation of the mechanism of action of various immunotherapy agents currently employed in clinical trials for Hürthle cell thyroid cancer.

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
