# Peer review of "The Molecular Landscape of Hürthle Cell Thyroid Cancer Is Associated with Altered Mitochondrial Function—A Comprehensive Review"

_cells, 2020, doi:10.3390/cells9071570_

Round 1

Reviewer 1 Report

Hürthle cell thyroid cancer is a rare thyroid cancer.  It is clinically characterized by invasiveness, and a poor responsiveness to radioactive iodine and chemotherapy. In addition, the cellular and molecular phenotype of Hürthle cancer cells has notable features. This justifies studies and reports in this field.

This paper by Kumari et al aims to offer a “comprehensive literature review on the association between the molecular features of HTC and cancer metabolism

Indeed, the report of the scientific literature seems exhaustive, but some improvements are required to offer a more useful paper to the future readers.

Major criticisms

  • In a review, the readers expect something other in addition to a simple and systematic report of data. The authors should include their expert opinions about some relevant related issues (for exampe extrapolations of the review's findings to clinical practice or what else we should study in future).
  • It should extremely useful to attempt to integrate the findings of the literature (genomic data, signaling alterations, metabolic profile) to obtain a signature which allows to identify clinically aggressive forms versus low risk HT.
  • In my opinion what is presented in the “Conclusion” is only a summary of the previous paragraph 10 and does not reflect the main issue of the review which is focused on “The Molecular Landscape of Hürthle Cell Thyroid”. I think that a significant rewriting is required.

Minor criticisms

Paragraph 2.

Since this paper is a review and not a statistic meta-analysis of published studies, this paragraph describing “Search strategy and selection criteria” could be omitted.

Paragraphs 5 and 6.

These two paragraphs discuss in detail mtDNA mutations and altered mitochondrial functions. However, the authors should indicate which alterations are specifically (or near-specifically) harbored in HTC versus those present in many others tumor histotypes. Probably a table could be sufficient.

Paragraph 8. 

 In my opinion this paragraph is not appropriate since it describes a very general topic, FDG-PET/CT, a method used mostly in tumors. No HTC-specific issue is discussed. I suggest omitting this paragraph.

Paragraph 10.

As the authors state (there is a need for new therapeutic approaches targeting …), at moment we do not have a targeted therapy for HTC.  Hence my suggestion is to shorten the paragraph to avoid describing general issues (for exampe immunotherapy).

On the same line, the author could delete fig.2 using the available space to insert useful pictures/tables to clarify inherent aspects of the review’s topic.

Author Response

We appreciate the thoughtful review of our manuscript, “The Molecular Landscape of Hürthle Cell Thyroid Cancer Is Associated with Altered Mitochondrial Function – a Comprehensive Review”. The authors thank the reviewers for their constructive suggestions for the improvement of the manuscript. Please find below our detailed response to the reviewers and editors’ feedback along with the revised manuscript with the changes captured in the track changes mode. Thank you for considering our submission for publication in Cells.

REPLY TO REVIEWER’S COMMENTS

Reviewer #1:  

Hürthle cell thyroid cancer is a rare thyroid cancer.  It is clinically characterized by invasiveness, and a poor responsiveness to radioactive iodine and chemotherapy. In addition, the cellular and molecular phenotype of Hürthle cancer cells has notable features. This justifies studies and reports in this field.

This paper by Kumari et al aims to offer a “comprehensive literature review on the association between the molecular features of HTC and cancer metabolism

Indeed, the report of the scientific literature seems exhaustive, but some improvements are required to offer a more useful paper to the future readers.

Question 1: In a review, the readers expect something other in addition to a simple and systematic report of data. The authors should include their expert opinions about some relevant related issues (for example extrapolations of the review's findings to clinical practice or what else we should study in future).

Answer 1: We agree with the Reviewer’s comments and have now made appropriate changes in the conclusion part of the revised manuscript page 7-8, lines: 303-319.

“HTC is distinct from other cancers based on extensive mtDNA mutations, as well as whole chromosome losses. These genetic alterations lead to decreased oxidative phosphorylation, enhanced aerobic glycolysis and oxidative stress as well as overactivation of PI3K/AKT/mTOR and RAS/RAF/MEK/ERK signaling pathways. Moreover, high-risk HTC is characterized by a significant mutation burden, and aneuploidy which may result in an enhanced immunogenicity of the tumor (Wang et al., 2019), and consequently better response to immunotherapy. This distinct molecular landscape may form a basis for novel therapeutic approaches involving combination therapies affecting overactive signaling pathways, cancer metabolism and immune landscape, as currently available treatment with radioactive iodine and FDA-approved TKIs is not effective. It is worthwhile to speculate that targeting overactive signaling pathways with TKIs and/or mTOR inhibitors in combination with small molecules blocking glucose transporters GLUTs or enzymes involved in overactive aerobic glycolysis such as hexokinase inhibitors, along with targeting the tumor microenvironment by immunotherapy, might be an effective strategy for novel clinical trials. HTC therapy is an evolving field and the recent findings described in this review may form a basis to identify effective strategies which can be exploited to prolong the survival rate of patients with HTC.”

Question 2: It should extremely useful to attempt to integrate the findings of the literature (genomic data, signaling alterations, metabolic profile) to obtain a signature which allows to identify clinically aggressive forms versus low risk HT.

Answer 2: We appreciate the Reviewer’s suggestions and have now included the details of the important differentiating features between low risk and high risk HTC in the introduction part (page 2, lines: 69-74) of the revised manuscript.

“HTC is further characterized based on the extent of vascular invasion – tumors with < 4 foci of vascular invasion are categorized as minimally invasive while tumors with ≥ 4 foci are categorized as widely invasive. Moreover, low-risk HTC is characterized by a different molecular profile compared with high-risk HTC, with the latter characterized by a significantly higher mutation burden, frequent loss of heterozygosity and mitochondrial DNA mutations affecting complex I involved in the electron transport chain [8, 9].”

Question 3: In my opinion what is presented in the “Conclusion” is only a summary of the previous paragraph 10 and does not reflect the main issue of the review which is focused on “The Molecular Landscape of Hürthle Cell Thyroid”. I think that a significant rewriting is required.

Answer 3: We agree with the Reviewer’s comments and have now significantly modified the conclusion part – page 7-8, Section 9 (Summary and Conclusions), lines 303-319, and above (response to question 1)  Question 4: Since this paper is a review and not a statistic meta-analysis of published studies, this paragraph describing “Search strategy and selection criteria” could be omitted. Answer 4: We appreciate the Reviewer’s suggestions; however, a comprehensive review may benefit from inclusion of the search strategy. With Reviewer’s kind permission, we kept this paragraph as informative for the readers.  

Question 5: Paragraphs 5 and 6 discuss in detail mtDNA mutations and altered mitochondrial functions. However, the authors should indicate which alterations are specifically (or near-specifically) harbored in HTC versus those present in many others tumor histotypes. Probably a table could be sufficient.

Answer 5: We have now included the details of the specific/near-specific alterations in complex I of HTC compared to the other tumor types in chapter 6 of the revised manuscript – page 5, line 193-198.

“The complex I mutations can occur in ND1, ND2, ND3, ND4, ND5, ND6 and ND4L regions of the mitochondrial genome [10]. HTC harbors the near-specific aberrations in ND2 and ND4 complex I regions. In contrast, mutations in ND1, ND3, ND5, ND6 and ND4L region of complex I have been found in other tumors such as prostate cancer [11], pancreatic cancer [12], colon cancer [13, 14], [15]. bladder cancer [16],breast cancer [17], and medulloblastoma [18].”

Question 6:  In my opinion this paragraph is not appropriate since it describes a very general topic, FDG-PET/CT, a method used mostly in tumors. No HTC-specific issue is discussed. I suggest omitting this paragraph.

Answer 6: We agree with the Reviewer’s comments. As it has been mentioned that the FDG-PET/CT scanning is significant for detection of HTC metastases and is very sensitive compared to the other detection techniques, we have now integrated chapters 7 and 8 to prevent confusion, and also to fulfil the suggestions by the other reviewers.  

Question 7:  As the authors state (there is a need for new therapeutic approaches targeting …), at moment we do not have a targeted therapy for HTC.  Hence my suggestion is to shorten the paragraph to avoid describing general issues (for example immunotherapy).

On the same line, the author could delete fig.2 using the available space to insert useful pictures/tables to clarify inherent aspects of the review’s topic.

Answer 7: We appreciate the Reviewer’s comments. The goal of mentioning the details about immunotherapy and other classes of medications was to present information about the current advances in terms of ongoing clinical trials including HTC patients. Describing immunotherapy might be particularly important in view of the fact that high risk HTC is characterized by a significant mutation burden and tumor aneuploidy, which is often associated with enhanced immunogenicity, and as such good response to immunotherapy (Wang et al., 2019). https://www.ncbi.nlm.nih.gov/pmc/articles/PMC6879305/pdf/elife-49020.pdf). Therefore, with the Reviewer’s permission, we would like to keep Figure 2, as it describes the agents used in the ongoing clinical trials including patients with HTC. For further clarification we changed the Figure 2 legend to: “Figure 2. Model depicting the drug targets in Hürthle cell thyroid cancer exploited in the ongoing clinical trials.” 

Reviewer 2 Report

As stated in the title, the authors of this manuscript aimed at writing a comprehensive literature review of Hürthle cell thyroid cancer molecular landscape. However, they did not manage to entirely fulfill this goal. Although it is well written and clear, the manuscript is too concise and omissive.

Main objection I have is that there are many similarities to the review by Ganly and McFadden (DOI: 10.1089/thy.2019.0088), raising a question of whether this review really brings something new to the field.

Some minor points:

  • In the Chapter 5, “Metabolic profile of HTC”, there is no mention of HTC metabolism at all, only definitions of Warburg and reverse Warburg effect are given.
  • I don't think Chapter 8 is necessary, it could simply be the last part of the previous chapter.
  • I would suggest rearrangement of the chapters, Chapter 9 should follow Chapter 6, discussing mitochondrial DNA mutations in HTC.

Author Response

We appreciate the thoughtful review of our manuscript, “The Molecular Landscape of Hürthle Cell Thyroid Cancer Is Associated with Altered Mitochondrial Function – a Comprehensive Review”. The authors thank the reviewers for their constructive suggestions for the improvement of the manuscript. Please find below our detailed response to the reviewers and editors’ feedback along with the revised manuscript with the changes captured in the track changes mode. Thank you for considering our submission for publication in Cells.

REPLY TO REVIEWER’S COMMENTS

Reviewer #2:

As stated in the title, the authors of this manuscript aimed at writing a comprehensive literature review of Hürthle cell thyroid cancer molecular landscape. However, they did not manage to entirely fulfill this goal. Although it is well written and clear, the manuscript is too concise and omissive.

Main objection I have is that there are many similarities to the review by Ganly and McFadden (DOI: 10.1089/thy.2019.0088), raising a question of whether this review really brings something new to the field.

Answer – We appreciate Reviewer’s constructive criticism. We agree that molecular landscape of Hürthle cell thyroid cancer was comprehensively captured by Ganly’s et al excellent review. That being said, current manuscript adds to the knowledge in the field by discussing potential clinical applications and translational perspective of the observed findings, summarizing some functional in vitro studies utilizing HTC models, underscoring the metabolic landscape of HTC and finally describing the ongoing clinical trials for HTC patients.

Question 1: In the Chapter 5, “Metabolic profile of HTC”, there is no mention of HTC metabolism at all, only definitions of Warburg and reverse Warburg effect are given.

Answer 1: We agree with the Reviewer’s comments and have now changed the title of chapter 5 to “Distinct metabolic profile of cancer cells”.

Question 2: I don't think Chapter 8 is necessary, it could simply be the last part of the previous chapter.

Answer 2: As per Reviewer’s suggestions, we have merged chapters 7 and 8.

Question 3: I would suggest rearrangement of the chapters, Chapter 9 should follow Chapter 6, discussing mitochondrial DNA mutations in HTC.

Answer 3:We agree with the Reviewer’s comments and have now rearranged the chapters as suggested above.

Reviewer 3 Report

In this paper Kumari et al., describe the genetic alterations in Hürthle-cell thyroid carcinoma (HTC) with the aim to provide a comprehensive review on the molecular landscape of HTC, particularly the widely invasive type. Although this aim renders the review potentially interesting, however careful reading does not reveal whereas and what is the difference between a benign thyroid Hürthle cell adenoma and an Hürthle cell carcinoma in terms of their molecular profile.

Major points

  • The careful reading reveals that all the informations described and discussed are generally related to HCTs without difference between a benign thyroid Hürthle cell adenoma and an Hürthle cell carcinoma in terms of their molecular profile.
  • In the section 5, References 40,41 and 42 are interesting reviews covering some aspects of cancer metabolism and how the metabolic rewiring hallmark may be targeted as an anticancer strategy. However, from these reviews and from the text reported (section 5), no information emerges regarding the specific metabolic profile of HTC that instead it discussed in section 6. Thus I would suggest to eliminate section 5 or address better its title.
  • In section 7, Authors reported that glycolysis upregulation is evident in HCT. Futher, the increase of GLUT1, MCT4 and CAIX are described in HCT. It is well known that the expression of these enzymes is regulated by HIF1a during tumorigenesis. In order to offer a complete vision on this topic to the reader, Authors should implement this section and discuss the role of HIF1a and how this transcription factor may modulate the metabolic adaptaion in HCT context in terms of glycolysis versus oxidative metabolism (TCA/OXPHOS and glutaminolysis).
  • As paragraph 8 reports the definition and diagnostic use of PET/CT scan mainly, I would suggest to integrate the text in paragraph 7.
  • Since Authors at the end of section 7 have stated that "The enhanced glucose uptake and increased glycolysis of HTC cells ..." it would be interesting for the reader to discuss the Reader the interplay between glycolysis and mitochondria/oxidative stress. Otherwise the findings reported in section 7 seem in contrast to what is reported in section 9 and this may generate misunderstanding.
  • In order to increase the scientific contribution of this Review, Authors should add and speculate on possible new perspectives and points of view in the HCT field.

Author Response

We appreciate the thoughtful review of our manuscript, “The Molecular Landscape of Hürthle Cell Thyroid Cancer Is Associated with Altered Mitochondrial Function – a Comprehensive Review”. The authors thank the reviewers for their constructive suggestions for the improvement of the manuscript. Please find below our detailed response to the reviewers and editors’ feedback along with the revised manuscript with the changes captured in the track changes mode. Thank you for considering our submission for publication in Cells.

REPLY TO REVIEWER’S COMMENTS

Reviewer #3:

In this paper Kumari et al., describe the genetic alterations in Hürthle-cell thyroid carcinoma (HTC) with the aim to provide a comprehensive review on the molecular landscape of HTC, particularly the widely invasive type. Although this aim renders the review potentially interesting, however careful reading does not reveal whereas and what is the difference between a benign thyroid Hürthle cell adenoma and an Hürthle cell carcinoma in terms of their molecular profile.

Question 1: The careful reading reveals that all the information described and discussed are generally related to HCTs without difference between a benign thyroid Hürthle cell adenoma and an Hürthle cell carcinoma in terms of their molecular profile.

Answer 1: We appreciate the Reviewer’s suggestions. We have included the key differences between a benign thyroid Hürthle cell adenoma and an Hürthle cell carcinoma in the introduction part, page 2 (lines 57-67) of the revised paper.

“The main distinction between the benign Hürthle cell adenoma and HTC s based on copy number alterations and nearly complete genome haploidization specific to HTC, as identified by ThyroSeq v3.0 [4]. Benign call rate is another measure to compare Hürthle cell adenomas and HTC using GSC, which combines next generation sequencing with machine learning tools. As a result, increasing number of indeterminate nodules have been determined as benign in nature. [5, 6]. Consequently, this has led to a significant improvement in specificity and positive predictive value of the test, with predicted ability to avoid up to 60% of unnecessary surgeries for benign conditions [8]. The remaining 40% is a result of the overlap between the benign and malignant Hürthle cell neoplasms that still exists and refers to common somatic mutations, such as variants of RAS gene, associated with both benign and malignant thyroid tumors.”

Question 2: In the section 5, References 40,41 and 42 are interesting reviews covering some aspects of cancer metabolism and how the metabolic rewiring hallmark may be targeted as an anticancer strategy. However, from these reviews and from the text reported (section 5), no information emerges regarding the specific metabolic profile of HTC that instead it discussed in section 6. Thus I would suggest to eliminate section 5 or address better its title.

Answer 2: We appreciate the Reviewer’s comments. We have now renamed the chapter 5 as “Distinct metabolic profile of cancer cells”.

Question 3: In section 7, Authors reported that glycolysis upregulation is evident in HCT. Further, the increase of GLUT1, MCT4 and CAIX are described in HCT. It is well known that the expression of these enzymes is regulated by HIF1a during tumorigenesis. In order to offer a complete vision on this topic to the reader, Authors should implement this section and discuss the role of HIF1a and how this transcription factor may modulate the metabolic adaptation in HCT context in terms of glycolysis versus oxidative metabolism (TCA/OXPHOS and glutaminolysis).

Answer 3:We are thankful for the Reviewer’s suggestions, and have now included the details about the role of HIF-1α in HTC tumorigenesis in the “Enhanced glycolysis in HTC” section, (page 6, lines 253-257 ). “The transcription factor HIF-1α (hypoxia inducible factor- 1α) is known to be strongly associated with increased glucose metabolism and angiogenesis in cancer [22-26]. Studies have reported expression of VEGF, a target gene of HIF-1α, in benign as well as malignant Hürthle-cell tumors, suggesting activation of HIF-1α pathway in these lesions [27]. So far, there is no concrete evidence about the direct role of HIF-1α in promoting HTC tumorigenesis.”

Question 4: As paragraph 8 reports the definition and diagnostic use of PET/CT scan mainly, I would suggest to integrate the text in paragraph 7.

Answer 4: We agree with the Reviewer’s suggestions and have now merged the paragraphs.  

Question 5: Since Authors at the end of section 7 have stated that "The enhanced glucose uptake and increased glycolysis of HTC cells ..." it would be interesting for the reader to discuss the Reader the interplay between glycolysis and mitochondria/oxidative stress. Otherwise the findings reported in section 7 seem in contrast to what is reported in section 9 and this may generate misunderstanding.

Answer 5: We have included the information regarding the interplay between glycolysis/mitochondria/oxidative stress (page 5-6, lines: 218-236).“ It is widely accepted that aberrant mitochondrial respiration and dysregulated mitochondrial function can result in increased oxidative stress, which can further lead to oncogenesis. Therefore, as the Hürthle cells have abundant mitochondria, there was a recent attempt to investigate the association between genes involved in the oxidative stress response and HTC. HTCs demonstrate increased production of reactive oxygen species (ROS), which enhances oxidative stress. These events further induce the oncogenic signaling pathways causing malignant transformation of cells and resistance to several therapeutic drugs and radiation therapies [6]. Moreover, NFE2L2 (nuclear factor erythroid 2-related factor 2) and KEAP1 (kelch-like ECH-associated protein 1) mutations have been identified in the samples derived from patients with HTC [5, 6]. These alterations might be of importance, as NFE2L2, which is negatively regulated by KEAP1, enhances survival after cellular damage [55-58]. Moreover, there are no significant changes in the anti-oxidative stress machinery in HTC. Krhin et al. showed no direct association between the expression of antioxidant genes including GPX1 (glutathione peroxidase 1), GSTP1 (glutathione-S-transferase P1), GSTT1 (glutathione-S-transferase T1), GSTM1 (glutathione-S-transferase M1), SOD2 (superoxide dismutase 2) and CAT (catalase), and the development of HTC. Interestingly, the study suggested that the GPX1 Pro198Leu polymorphism might be associated with HTC risk [54]. However, these findings need to be validated within a larger population. The dysregulated mitochondrial function leading to excessive ROS production and downregulated OXPHOS observed in HTC is associated with compensatory upregulation of aerobic glycolysis [19-21].”

Question 6: In order to increase the scientific contribution of this Review, Authors should add and speculate on possible new perspectives and points of view in the HCT field.

Answer 6: We appreciate the Reviewer’s comments. The novel perspectives in the HTC field have been discussed in the conclusion part (Section 9, page 7-8, lines 303-319) of the revised manuscript.

“HTC is distinct from other cancers based on extensive mtDNA mutations, as well as whole chromosome losses. These genetic alterations lead to decreased oxidative phosphorylation, enhanced aerobic glycolysis and oxidative stress as well as overactivation of PI3K/AKT/mTOR and RAS/RAF/MEK/ERK signaling pathways. Moreover, high-risk HTC is characterized by a significant mutation burden, and aneuploidy which may result in an enhanced immunogenicity of the tumor (Wang et al., 2019), and consequently better response to immunotherapy. This distinct molecular landscape may form a basis for novel therapeutic approaches involving combination therapies affecting overactive signaling pathways, cancer metabolism and immune landscape, as currently available treatment with radioactive iodine and FDA-approved TKIs is not effective. It is worthwhile to speculate that targeting overactive signaling pathways with TKIs and/or mTOR inhibitors in combination with small molecules blocking glucose transporters GLUTs or enzymes involved in overactive aerobic glycolysis such as hexokinase inhibitors, along with targeting the tumor microenvironment by immunotherapy, might be an effective strategy for novel clinical trials. HTC therapy is an evolving field and the recent findings described in this review may form a basis to identify effective strategies which can be exploited to prolong the survival rate of patients with HTC.”

Round 2

Reviewer 1 Report

In the present version, the authors addressed my concerns

Reviewer 3 Report

I appreciate the extensive revision that the authors made. The revised manuscript is improved and can be accepted for publication.